# Metallic Nanoparticles: Their Potential Role in Breast Cancer Immunotherapy via Trained Immunity Provocation

**DOI:** 10.3390/biomedicines11051245

**Published:** 2023-04-23

**Authors:** Elham Zarenezhad, Manal Hadi Ghaffoori Kanaan, Sura Saad Abdollah, Mohammad Kazem Vakil, Mahrokh Marzi, Abdulbaset Mazarzaei, Abdolmajid Ghasemian

**Affiliations:** 1Noncommunicable Diseases Research Center, Fasa University of Medical Sciences, Fasa 7461686688, Iran; 2Department of Agriculture, Technical Institute of Suwaria, Middle Technical University, Baghdad 9768876516, Iraq; 3Suwaria Primary Health Care Sector, Wassit Health Office, Sharjah 9668866516, Iraq; 4Department of Immunology, School of Medicine, Iranshahr University of Medical Sciences, Iranshahr 7618815676, Iran

**Keywords:** breast cancer, nanoparticles, trained immunity, cancer therapy, drug delivery

## Abstract

Owing to drawbacks in the current common cancer therapies including surgery, chemotherapy and radiotherapy, the development of more reliable, low toxic, cost-effective and specific approaches such as immunotherapy is crucial. Breast cancer is among the leading causes of morbidity and mortality with a developed anticancer resistance. Accordingly, we attempted to uncover the efficacy of metallic nanoparticles (MNPs)-based breast cancer immunotherapy emphasizing trained immunity provocation or innate immunity adaptation. Due to the immunosuppressive nature of the tumor microenvironment (TME) and the poor infiltration of immune cells, the potentiation of an immune response or direct combat is a goal employing NPs as a burgeoning field. During the recent decades, the adaptation of the innate immunity responses against infectious diseases and cancer has been recognized. Although the data is in a scarcity with regard to a trained immunity function in breast cancer cells’ elimination, this study introduced the potential of this arm of immunity adaptation using MNPs.

## 1. Background

Current cancer therapy approaches are associated with drawbacks in terms of their costs (e.g., the sociological and economic aspects), resistance, recurrence, toxicity and various side effects [1,2,3,4,5]. Cancer immunotherapy has attracted huge attention during the recent decades [6,7,8]. Cancer immunotherapy is a novel approach with several food and drug administration (FDA)-approved drugs [9]. Of these, immune checkpoint inhibitors (ICIs) are considerable mostly for use against soft tumors, while neoantigens are cancer-specific antigens with promising targets for immunotherapy. Tumor-infiltrating T-lymphocytes (TILs) can be provoked or engineered particularly against melanoma cells; however, a high load of interleukins 2 and 15 is a limitation leading to cytotoxicity [10]. The human immune system is capable of mounting its memory and adapting its responses to infections and cancer following exposure to various microorganisms and other immunogens [11,12]. It has been revealed that both adaptive and innate immune responses can be adapted to infections and vaccinations [13]. A de facto memory, exerted by an innate immunity following the recurrence of infection, was proposed as a “trained immunity” in 2011 [14]. Trained immunity was also earlier observed following a response to a *Candida albicans* infection in T and B cell-deficient mice which was conferred by monocytes and macrophages [15]. Vaccination following BCG has also induced a higher response of monocytes [16]. Moreover, trained immunity has also participated in anti-cancer and immunostimulatory effects, for example, counteracting the bone marrow function in developing immune-suppressive polarized cells [11,17] (Figure 1 and Figure 2).

Therefore, due to the immunosuppressive nature of the tumor microenvironment (TME) and the poor infiltration of immune cells, the potentiation of an immune response or direct combat is a goal employing nanoparticles (NPs) as a burgeoning field [18,19,20]. Various NPs-based cancer therapy platforms include liposomes, niosomes, polymeric NPs, micelles, cell-derived NPs, inorganic NPs and nanogels [5,21]. Trained immunity has been observed in various cancers such as bladder (BCG-induced) and head cancers, osteosarcoma (Muramyl peptide) and other cancer cells (Beta glucan) [22,23,24,25].

The other edge of sword involves the negative/deleterious effects of trained immunity which cause hyperinflammation, immune paralysis and atherosclerosis. Therefore, a NPs-based immunotherapy can inevitably reduce the risk of systemic toxicity and adverse events of immune stimulation [26,27]. NPs can carry immune compartments such as antibodies, anti-PD-1 drugs or siRNAs or directly activate the host responses, hence, developing a trained immunity or an immune tolerance [28,29]. Considering these, the future therapeutic choices of trained immunity should cover the side effects using combinations or alterations in the current approaches. Numerous innate immune stimulators have been reviewed [23]. For example, the induction of macrophages, NK cells and epithelial ILC2 cells leads to the secretion of tumor necrosis factor (TNF)-α, INF-γ, IL-5 and IL-33 cytokines [30,31]. The immunotherapy of cancer has demonstrated acceptable outcomes, though medical, sociological and ethical issues remain to be solved. NPs-based approaches have caused an intelligent response to the TME which has an immunosuppressive nature [32]. Moreover, NPs have recently accelerated the diagnosis and treatment of diseases which can be improved by mathematical modeling to adopt optimal efficient levels. In addition, univocal strategies should be replaced by combination therapies. Furthermore, an innovative integration of various agents in a nanoplatform significantly provokes immune responses. Consequently, the time-efficient, cost effectiveness and immune-stimulatory traits of NPs-based platforms are promising in the context of cancer therapies.

### 1.1. Trained Immunity Related Data in Breast Cancer Therapy Is in Scarcity

NPs have improved the anticancer effects of radiotherapy by stimulating the innate immunity via toll-like receptors (TLRs) 2, 3 and 4 [33,34,35,36]. CD163+ tumor-associated macrophages also have an outlined role [37]. It has been demonstrated that iron oxide NPs (FeONPs) can polarize macrophages into an M1 cancer-eliminating phenotype [38]. Another recent study used NPs (20 µg/mL)-incorporated stimulator of an interferons genes (STING)-agonist for the durable and efficient stimulation of the innate immunity via provoking the INF-β-mediated responses [39]. Additionally, pH sensitive Ferritin-based NPs (FPBC@SN) have also caused a T cells activation and cytokines release promoting their anticancer efficacy against breast cancer [40]. Clinical trials using a Clodronate, IPI-549 and IFN-γ application have exhibited macrophages activation against breast cancer (NCT00009945, NCT03961698, and NCT03112590). However, interleukin 6 (IL-6) and TNF can enhance tumor proliferation and metastasis which needs exact considerations [41]. According to various reports, an NPs-based breast cancer treatment is promising (Figure 1, Figure 2, Figure 3 and Figure 4). This paper was a systematic review of studies during 2012–2022 regarding the immunotherapy of breast cancer using metallic NPs for the treatment of breast cancer. This study was a systematic review covering a period of 10 years. NPs and the trained immunity mechanisms of anticancer activity, and also nano-carriers were reviewed. The literature search was performed in Scopus, Google Scholar, PubMed, Web of Science and Google using the keywords: Metallic nanoparticles ∗ AND trained immunity, Metallic nanoparticles ∗ AND Breast Cancer, Metallic nanoparticles ∗ AND Breast cancer stem cells, and Immunomagnetic nanoparticles ∗ AND Breast cancer. The results were screened based on the title, abstract, and full text availability (Figure 3).

### 1.2. The Immunotherapy of Cancers

Although various therapies have been available for cancer, a resistance, low response or limited efficacy, off-target toxicity and high costs are the main drawbacks of traditional therapies (e.g., radiotherapy and chemotherapy). Immunotherapy boosts a host body’s defense against cancer cells specifically. Traditional cancer therapies have not considered immune suppression in the TME, but various immunotherapies are in use such as checkpoint inhibitors, cell therapies, chimeric antigen receptor (CAR)-based cell therapies, and oncolytic viruses that require efficient T cell responses to provoke a durable clinical response [42]. Several FDA-approved immunotherapy drugs include Ipilimumab, Pembrolizumab, Nivolumab, Atezolizumab, Avelumab and Durvalumab used as monoclonal antibodies [43].

## 2. Trained Immunity Cells and Compartments

Innate immune cells include phagocytes (neutrophils, macrophages, innate lymphoid cells (ILCs), monocytes, basophils and eosinophils) and natural killer (NK) cells that have originated from bone marrow. These cells react as the prompts and non-specific first-line of responses and activate adaptive immunity. Alterations mainly of the metabolic and epigenetic aspects render an innate immune memory or trained immunity. Those bone marrow progenitor cells and hematopoietic stem cells regulate that innate immunity. Recognizing the components of innate immunity can be stimulated by nanoparticles such as dendritic cells (DCs) surface PRRs which lead to the activation of various signaling pathways and type I interferon (IFN-I). This phenomenon activates the DCs’ and T cells’ responses and shifts a “cold” tumor to a “hot” tumor, enhancing the efficiency of the immune checkpoint inhibitor treatment of cancer. The activation of M2 macrophages and the stimulator of interferon response (STING) has been demonstrated using nanomedicine [44,45,46,47,48]. Moreover, BCG-mediated trained NK cells were shown to release VEGF-α and IFN-γ [48]. Additionally, the epigenetic and transcriptomic provoking of trained neutrophils has been also deciphered [49].

## 3. Metallic Nanoparticles and the Immunotherapy of Cancers

The efficacy and efficiency of MNPs is substantially influenced by their size, shape and surface characteristics. For example, AuNPs and AgNPs are widely used. Various biomedical and pharmacological MNPs include AuNPs, AgNPs, CuNPs, FeNPs, NiNPs, TiO_2_NPs, SiNPs and FeONPs. Due to a higher density, their absorption by cells is easier, thus, they can deliver drugs/biomolecules more conveniently and rapidly. The anticancer mechanisms of MNPs mostly include a cell membrane disturbance and a protein denaturation, the inhibition of DNA replication, the transcription of genes, inactivation of enzymes, signal transduction and their binding to sulfur and phosphorus ions as the major compartments of cellular macromolecules [50,51,52,53]. Nanocarriers increase the bioavailability, solubility, local concentration, TME targeting improvement and cell absorption of drugs, particularly through vectorization [4,54,55]. Those tumor-specific antigens (TSAs) can be efficiently transferred to lymph nodes using NPs depending on their physical characters such as the surface and size [56]. Noticeably, positively charged NPs induce higher immune responses despite having a lower penetration [57]. NPs with a suitable size (e.g., 25 nm) also protect or encapsulate and transfer the TSAs into the TME to induce immune cells. Therefore, a local cancer therapy using NPs and immunotherapy confers promising effects in terms of its lacking a systemic toxicity [58]. Furthermore, NPs have been applied in cell therapy, TME immune induction and for provoking innate immune cells. It is worth mentioning that a combination of long-term vaccinations and short-term nanomaterials-based cancer therapies contributes to efficient outcomes [59,60]. PLGA nanoparticles have been used for the delivery of antigens and vaccines to innate immune cells [60,61]. For example, pH-sensitive liposomes have been applied for the delivery of proteins to DCs in cancer-recruiting CD8-positive T cells. Additionally, micelles can transfer AuNPs to the cytoplasm for a better delivery of antigens [62,63]. AuNPs have been used for the delivery of adjuvants or various proteins, genetic drugs, and immune antibodies [63]. More relevantly, NPs carry specific antigens to the TME, and counteract the aberrant behavior of cells (e.g., mitigating drug resistance) [64]. However, MNPs can accumulate in the body and various methods are available for their metabolization and removal such as through normal excretion, using enzymes, microorganisms, nano-architectures, and MNPs’ PEGylation, zwitterionization, and precoating [65,66]. The normal excretion of MNPs depends on their size and surface charge [67]. Renal and hepatic pathways also remove NPs from the body. Gold NPs (AuNPs) have demonstrated theranostic anticancer traits in various sizes and shapes, while the green synthesis of AuNPs is promising regarding lower costs and pollution [68]. PEGylated porphyrins silver NPs (AgNPs), namely, a AgNP@P(PEG350)3 hybrid system, was also shown to be promising as a theranostic tool [69].

## 4. Metallic Nanoparticles for the Immunotherapy of Breast Cancer

Molybdenum nanosheets or 2D transition metal dichalcogenides (TMDs, MoS_2_) could stimulate a trained immunity via macrophages’ activation following a subsequent exposure to microbial ingredients in the epigenetic (e.g., histone methyl transferase) and metabolic (e.g., glycolysis via cyclic adenosine monophosphate, cAMP) pathways [70], and an IL-6, TNF-α and an IL-1β and TGF-β superfamily proinflammatory cytokines release [71,72,73]. In a study by Shi X. et al., exosome (epidermal growth factor receptor 2 (HER2) and anti-human CD3)-producing cells which excreted engineered, cell-derived NPs has been used for targeted breast cancer therapy [74]. A combination therapy using immunotherapy (poly-L histidine), chemotherapy (doxorubicin) and nanoparticles-stimulated macrophages, exerted breast cancer death [75]. Moreover, the application of a bimetallic metal-organic framework (MOF) nanoparticle, which involved Gd^3+^ and Zn^2+^ (Gd-MOF-5), could stimulate immune responses and overcome the suppressive effects of phosphatidyl serine (PS) [76].

Kievit, Forrest M. et al. [77] described the rational expansion and application of multifunctional superparamagnetic iron oxide NPs (SPIONs) to target metastatic breast cancer (MBC) in a mouse model and magnetic resonance imaging (MRI). Super-paramagnetic FeONPs covered with a chitosan and polyethylene glycol (PEG) were named with a fluorescent color for optical discovery and interlaced with a monoclonal antibody versus the neu receptor (NP-neu). A NP-neu can mark early breast tumors and remarkably, just a NP-neu bound to the automatical lung, liver, and bone marrow metastases in a transgenic mouse pattern of MBC, highlighting the urgency of targeting for a delivery to metastatic disease. The SPIONs supplied a remarkable contrast increase in the MRI of prime breast tumors; therefore, they show potential for discovering MRI micrometastases and for preparing a great platform for the further expansion of an effective treatment for MBC.

Li, Fu-Rong et al. [78] provided magnetic NPs with carbon-coated pure iron (Fe@C) as a core and functionalized with an epithelial cell adhesion molecule (EpCAM) monoclonal antibody for immunomagnetic NPs (IMPs). The IMPs combined with immunocytochemistry (ICC) were used to develop an IMP-enriched assay to identify circulating tumor cells (CTCs). This procedure reached an identical sensitivity, but with a remarkably decreased false-positive speed (compared with a nested RT-PCR). This method will assist in finding occult micrometastases, erect clinical staging and a conductor-individualized therapy after surgery, accentuating a considerable potential valence in the clinic.

Paholak, Hayley J. et al. [79] performed classical cancer stem cell (CSC) experiments, both in vitro and in vivo (immunodeficient mice), to outline the phototherapy of a highly crystalline IONs efficiency for killing breast cancer stem cells (BCSCs) and triple negative breast cancer (TNBC). Photothermal therapy (PTT) prevents BCSC self-renewal by reducing the mammosphere constitution in primary and secondary generations. In this study, the transduction potential of PTT was investigated using metastatic and immunocompetent mouse patterns. The PTT not only decreased the BCSCs but also reduced the metastasis to the lymph nodes and lung. The PTT combination therapies with standard approaches such as surgery can eradicate BCSC and cease metastasis, offering the long-term survival of patients with MBC.

Duan, Xiaopin et al. [80] revealed that Zn-pyrophosphate (ZnP) NPs loaded with photosensitizing pyrolipid (ZnP@pyro) induced necrosis and/or apoptosis and indirectly killed tumor cells under a light irradiation by interrupting the tumor vessels and enhancing the tumor immunogenicity. ZnP@pyro photodynamic therapy (PDT) sensitizes tumors to a PD-L1 antibody-mediated checkpoint inhibition (destroying the primary 4T1 breast tumor with an arrest of the lung metastasis). The ZnP@pyro PDT treatment added to anti-PD-L1 provoked cytotoxic T cell responses eradicating the photo-irradiated tumors. These results revealed that a nanoparticle-mediated photodynamic treatment can enhance the systemic effect of checkpoint blockade immunotherapies through activating the intrinsic and adaptive immune networks in the TME.

Ultimo, Amelia et al. [81] developed a toll-like receptor 3 (TLR3) delivery method relying on mesoporous silica NPs capped with synthetic twofold-stranded polyinosinic-polycytidylic acid (poly(I:C)) RNA for a controlled burden transfer in SK-BR-3 BC cells. The findings displayed that the poly(I:C)-conjugated NPs were effectively aimed at breast cancer cells (owing to a dsRNA/TLR3 interaction). Such an interaction induces the apoptotic paths in SK-BR-3, notably reducing the cell viability. The Poly(I:C) cytotoxic efficacy was increased in breast carcinoma cells using the filling of NPs mesopores with the anthracyclinic antibiotic doxorubicin, which is a usual chemotherapeutic factor.

Cheng, Ning et al. [82] applied liposomal nanoparticle-delivered cGAMP (cGAMP-NP) to induce the stimulator of IFN genes more successfully than soluble cGAMP. The cGAMP-NPs decreased the melanoma tumor burden (with a restricted responsiveness to anti-PD-L1) and impelled intrinsic and adaptive host-protected responses to existing tumors in orthotopic and genetically engineered patterns of basal-like TNBC. In the TME, the cGAMP-NPs interact with murine and human macrophages (M) to reprogram from a protumorigenic M2-like phenotype to an M1-like phenotype. The depletion of activated T cells reduces the antitumor activity. The cGAMP-NPs inhibit the foundation of peripheral tumors, and one dose is enough to prevent TNBC. This information showed that a minimum system containing cGAMP-NPs alone is adequate to adjust the tumor microenvironment to effectually control PD-L1–insensitive TNBC.

In another study, a TME-activated binary cooperative prodrug nanoparticle (BCPN) was drawn to reclaim immunotherapy by adjusting the immune tumor microenvironment synergy. The BCPN, consisting of tumor acidity and dual-responsive oxaliplatin (OXA), reduced the prodrug-mediated immunogenic cell death and active homodimer of NLG919 to inactivate indoleamine 2,3-dioxygenase 1 (participating in the TME immunosuppression), and also induced antitumor immunity. After the cleavage of the poly (ethylene glycol) shell caused by the tumor acidity, the prodrug nanoparticle exhibited a negative-to-positive charge switch to increase the tumor agglomeration and deep influence. OXA and NLG919 are activated via a glutathione-mediated reduction. According to the experiments, activated OXA increases the intratumoral agglomeration of cytotoxic T lymphocytes eliminating cancer cells. Then, NLG919 attenuates an IDO-1-mediated immunosuppression and represses regulatory T cells. The prodrug nanoparticle delivered substantially superior anticancer effects compared to free OXA or a combination of free OXA and NLG919, and also inhibited metastasis in colorectal and breast cancer mouse models [83].

Elsewhere, PEGylated bilirubin NPs (BRNPs) responsive to reactive oxygen species (ROS) were developed to enclose two glutathione-activating drugs, namely, dimer-7-ethyl-10 hydroxycamptothecin (d-SN38) and dimer-lonidamine (d-LND). Drug dimerization notably raised the drug loading valence and nanoparticle encapsulation performance. With the help of an iRGD peptide (cRGDKGPDC), the cellular uptake of the bilirubin NPs was more than double that of the controls. Then, the d-SN38 and d-LND were released from the NPs (SL@BRNPs) rapidly in the presence of intracellular ROS. In addition, pharmacodynamic experiments confirmed that the combination of SL@BRNPs with an anti-PD-L1 antibody greatly prevented primary BC cells, improving the CD8 + T cell surface area and CD8+ T cells/Tregs proportions in a tumor. In addition, it displayed a high immune memory efficacy and could inhibit the growth of lung metastasis. Overall, this strategy offers a modern procedure to rationally design nanocomposites through a combination of active drug dimers and the release of stimulus-responsive drugs, and it offers the successful use of new drug delivery systems in combination with antibody-blocking safety checkpoints [84,85].

Xu, Chenfeng et al. [86] reported a synergistic TNBC immunotherapy based on the induction and enhancement of an immune cell death (ICD) response through an active, nanoparticle-convertible approach for the simultaneous delivery of cisplatin, adjudin, and WKYMVm (or Trp-Lys-Tyr-Met-Val-D-Met hexapeptide). To reach a structural alteration with the benefits of optimal resizing, an efficient drug delivery, and a well-controlled distribution, the NPs could repeatedly respond to matrix metalloproteinases-2, glutathione, and the pH. The cisplatin and adjudins could synergistically enhance the cascade of ROS and ultimately increase the formation of highly toxic downstream hydrogen peroxide and .OH molecules, initiating the ICD response by endoplasmic reticulum stress mechanisms, cell apoptosis and autophagy. The WKYMVm could enhance the anti-TNBC immunity by activating the Formyl 1 peptide receptor to induce stable interactions between the dying cancer cells and dendritic cells. Therefore, NPs can achieve a primary tumor regression and inhibit metastasis, providing a significant survival advantage, by enhancing anti-TNBC immune responses.

An immune nanoparticle was constructed with cyclic diguanylate monophosphate (cdGMP), an interferon gene-stimulating pathway agonist, and monophosphoryl lipid A (MPLA), a toll-like receptor 4 (TLR4) agonist, synergized to generate high levels of Type I interferon β. Using a mouse pattern of metastatic TNBC, the systemic delivery of these immune NPs led to remarkable therapeutic results for an increase in antigen-presenting cells (APCs) and NK cells, comparably. These findings suggest that NPs can simplify the systemic delivery of multiple immunomodulatory cargoes for APC-based local and systemic antitumor immunity [87].

Castro, Flávia et al. [88] deciphered chitosan/poly (γ-glutamic acid) NPs’ (Ch/γ-PGA NPs) substantial stimulatory effects in combination with RT to induce antitumor immunity in a mouse model of 4T1 orthopedic breast tumor. Untreated animals had advanced primitive tumor growth and spleen and pulmonary metastases. While RT reduced the primary tumor burden, a treatment of Ch/γ-PGA NPs reduced the systemic immunosuppression and lung metastases. A combination treatment (i.e., RT + Ch/γ-PGA NPs) synergistically disrupted a 4T1 tumor improvement, which resulted in a considerable early tumor growth and splenic depletion, decreased the percentage of splenic immune-suppressive myeloid cells, and increased the anti-tumor CD4^+^IFN-γ^+^ population. The animals from the combination treatment showed smaller lung metastatic foci and lower areas of systemic cytokines of the IL-3, IL-4, IL-10, and CCL4 chemokines compared to the untreated animals.

Feng, Bing et al. [89] displayed the preparation of NPs for the self-assembly of small molecular drugs with an indocyanine green (ICG) pattern. The self-assembly of paclitaxel NPs (PTX) with an ICG pattern and the application of ISPN for a combined TNBC immunotherapy were revealed. The ISPN demonstrated a satisfactory colloidal stability and high efficiency for an ICG and PTX tumor-specific code delivery by increasing the tumor permeability and persistence effect. The ICG component of the ISPN successfully induced tumor cell death by activating an antitumor immune response via a photodynamic treatment; however, the PTX provided by the ISPN suppressed regulative T lymphocytes (T_regs_) to fight the immunosuppressive tumor microenvironment. A combined treatment of TNBC with ISPN and the therapy of a safe blockade with αPD-L1 showed a synergistic result on tumor regression, the prevention of metastasis and a debarment of recurrence.

Lei, Jun et al. [90] described the combined use of avasimibe and a safe pH-sensitive nanodrug delivery system composed of doxorubicin (DOX) and metal-organic framework NPs (MNPs). The results showed that the DOX-MNPs therapy prevented tumor growth with a good immunological profile, and the combined treatment with avasimibe DOX-MNPs treatment showed a better efficacy than monotherapies in the treatment of 4T1 breast cancer.

A thermo-sensitive PDLLA-PEG-PDLLA (PLEL) hydrogel-based topical, injectable, medicinal platform with a near-infrared (NIR)-stimulated medicine liberation was expanded to attain immunotherapy to prohibit breast cancer relapse. Self-assembled multifunctional NPs were combined with the thermosensitive PLEL hydrogel. Near-infrared induced the release of CPG ODNs and R848 immune components from the thermoresponsive PLEL hydrogels and these acted as cancer vaccines for post-surgical immunotherapy with an effective and sustained antitumor effect [91].

In this study, for the use of ferritinophagy-cascade ferroptosis and tumor immunity activation for cancer treatment, effective tumor targeting NPs comprising of ferritin and a pH-sensitive molecular-switch (FPBC@SN) were expanded. The cleavage of FPBC@SN in the acidic cytoplasm, produced sorafenib (SRF) and the indoleamine-2,3-dioxygenase (IDO) inhibitor (NLG919). The SRF increased the nuclear receptor coactivator 4 (NCOA4) to impel ferritin and to reduce the iron pool. Afterwards, the created iron ions took part in the Fenton reaction to generate lipid peroxide (LPO). However, sorafenib blocks glutathione synthesis to inhibit glutathione peroxidase 4 (GPX4), which can clear lipid peroxide as a diverse route from ferritinphagy to boost ferroptosis in tumor cells. NLG919 prevents indoleamine-2,3-dioxygenase from decreasing tryptophan metabolism, thus, anti-tumor immunity is stimulated. According to in vitro and in vivo experiments, the FPBC@SN prevented the growth and metastasis of tumor cells, indicating the potential of FPBC@SN, exerted based on a ferroptosis ferritinophagy-cascade combination and tumor immune activation [40].

Hu, Chuan et al. [92] described a new type of self-delivered NPs (MA-pepA-Ce6 NPs) for photosensitizing (chlorin e6, Ce6) and inhibiting programmed cell death ligand 1 (PD-L1). Simple and biodegradable substances offer a controllable production potential. In addition, MA-pepA-Ce6 NPs were reduced using matrix metalloproteinase-2 (MMP-2) in the TME and revealed VRGDK-Ce6. The exposed VRGDK-Ce6 exhibited a higher targeting ability to the α_v_β_3_ integrin receptor, ensuring enough agglomeration and strong laser-activated antitumor immunoregulatory effects. Notably, released metformin (MET) disrupted PD-L1 expression in the TME and enhanced a photodynamic therapy-induced antitumor immune response, thus, remarkably improving the treatment efficacy. In general, this study presents a potentially attractive paradigm of a PDT-induced synergistic immunotherapy using the revealing MET-mediated downregulation of PD-L1 to achieve tumor eradication.

Zhang, Yining et al. [93] investigated the hypothesis that intravenously administered, uncarried poly(DL-Lactide-co-Glycolide) (PLG) NPs could divert circulating immune cells from the TME to increase the effect of anti-PD-1 immunotherapy in a 4T1 mouse model of metastatic TNBC. Based on in vitro studies, these NPs reduced the expression of MCP-1 to five-fold and increased the expression of TNF-α to more than two-fold after the uptake using inherent, protected cells. An intravenous injection of particles led to internalization using myeloid-derived suppressor cells and monocytes. The nanoparticle delivery reduced numerous myeloid-derived inhibitor cells. The NPs, importantly, reduced the tumor growth and led to a survival advantage (in combination with an anti-PD-1 antibody). The gene expression resolution by a gene set enrichment analysis (GSEA) showed that the inflammatory myeloid cell paths were upregulated in the TME. Involvement in the extrinsic apoptotic paths was considered in the early tumor. Accordingly, the PLG NPs altered the responses and the TME profile to dominate the local immune suppression, mainly of myeloid cells, and to increase the effect the of anti-PD-1 treatment.

In this study, a 60 nm nanoparticle was loaded with a STING agonist, which induced the potent production of interferon ß, resulting in the activation of antigen-presenting cells (APCs). Moreover, mainstream ligands have also been used to target fibronectin, integrin α_v_ß_3_, and P-selectin, which are commonly used to guide NPs to tumors. From a 4T1 mouse model, the micro-distribution of four types of NPs in the tumor immune microenvironment was evaluated in three various perspectives of breast cancer. Different types of NPs using immune cell subsets led to a variable uptake depending on the organ and tumor stage. Among the types of NPs, therapeutic studies have showed that non-targeted NPs and integrin-targeted NPs display significant short-term and long-term immune responses and long-term antitumor efficacy [39].

Wu, Jinxian et al. [94] used glycated chitosan (GC) to fabricate a self-assembled GC@ICG NP available to tumor cells for a synergistic cancer treatment based on a combination of immunotherapy and phototherapy. The self-assembled synthesis of the globular GC@ICG notably improved the resistance of ICG and conferred the GC with Trojan Horses in the tumor cells to raise the tumor immunogenicity. A 4T1 bilateral tumor-bearing mouse model was created to appraise the therapeutic results and the host-specific antitumor immune response. In general, the GC@ICG-based phototherapy could destroy early tumors and resist the advancement of untreated distant tumors. Furthermore, the GC@ICG-based phototherapy was observed to repress lung metastasis and increase a CD8^+^ T cell infiltration in untreated distant tumors. Thus, this plan is promising in addressing the challenges of treating TNBC.

According to the results, Figure 4 summarizes the main findings of the NPs-based breast cancer immunotherapies focusing on a trained immunity.

## 5. Future Prospects

The exact appreciation of the trained immunity mechanisms and gaps in the knowledge is crucial to develop related immunotherapies. The determination of the functionality of genetic alterations is essential [95]. The green synthesis of MNPs, for example, will be promising for lowering the costs and pollution and for energy saving; however, their toxicity against normal cells should also be assessed. The application of combined therapies will also be helpful for more efficient cancer therapies [50]. The performance of clinical trials for the determination of a trained immunity anticancer efficiency combined with MNPs will open up avenues for cancer therapy. Future studies should address the question of NPs mechanisms for TME regulation and the induction of immunity. Furthermore, the proper formulations and combinations of MNPs with a suitable size and natural or synthetic drugs contribute to higher efficient cancer cell eradication by reducing off-target or systemic toxicity. A decrease in NPs accumulation in the body from a deeper knowledge of the proteins–MNPs interactions, and the application of suitably-sized and charged MNPs alongside further processes to mitigate a protein corona formation should also be taken into account [66].

## 6. Conclusions

Due to various drawbacks in the common cancer therapy approaches, such as off-target toxicity, the costs and non-adherence by all patients, immunotherapy has attracted huge attention during recent decades. It has been revealed that both adaptive and innate immune responses can adapt to infections and vaccinations. The activation and adaptation of innate immunity is promising compared to acquired immunity for rapid and extensive responses and lower side effects. However, considering the risk of systemic toxicity following immune cells’ stimulation, the employment of alternative approaches such as NPs’ formulation for specific anticancer stimulation or therapy is promising.

## Figures and Tables

**Figure 1 biomedicines-11-01245-f001:**
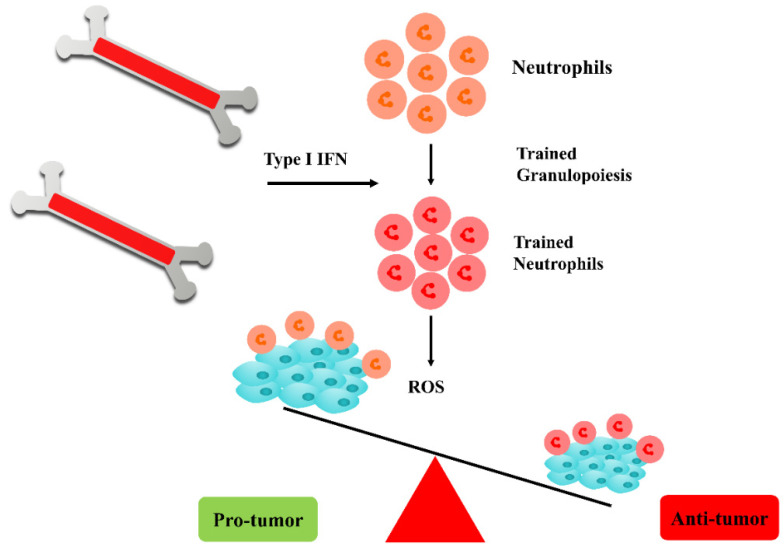
Trained neutrophils exerting anticancer effects; following cytogenesis in the bone marrow, these cells can be induced by type I interferons and proliferate to produce ROS free radicals to combat tumor cells. INF: interferon, ROS: reactive oxygen species.

**Figure 2 biomedicines-11-01245-f002:**
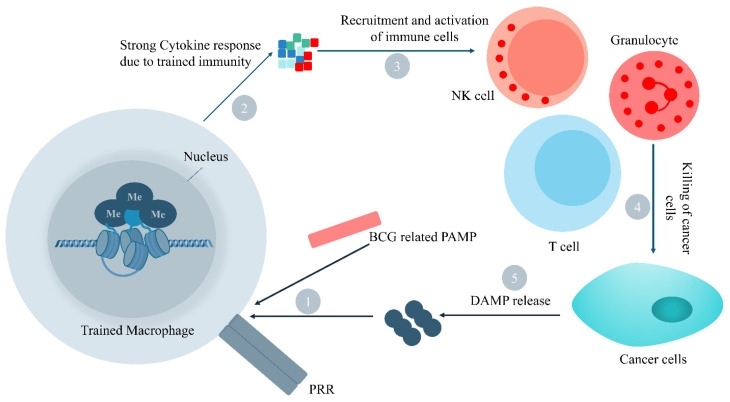
Trained macrophage which confers anticancer effects following BCG vaccination; macrophages exposed to BCG- or pathogen-associated PAMPs or DAMPs produce and secrete cytokines massively which recruit other immune cells to kill cancer cells; NK: natural killer, DAMP: damage associated molecular patterns, PAMPs: pathogen associated molecular patterns, PRR: pathogen recognition receptor.

**Figure 3 biomedicines-11-01245-f003:**
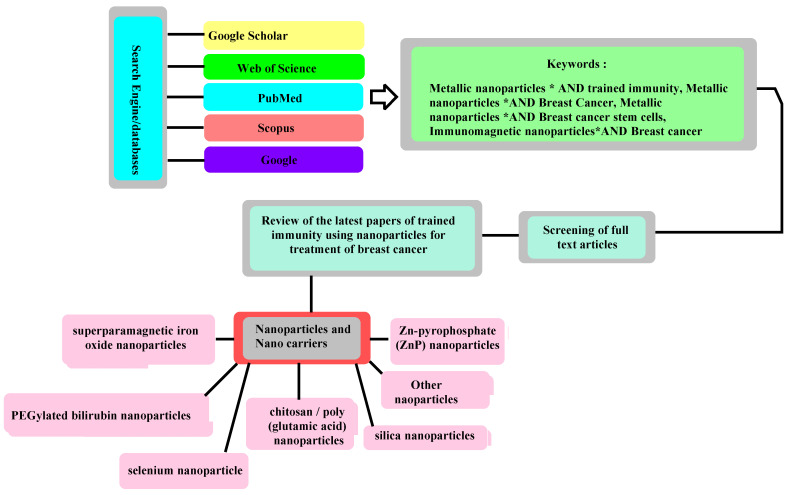
The work-flow of this review study.

**Figure 4 biomedicines-11-01245-f004:**
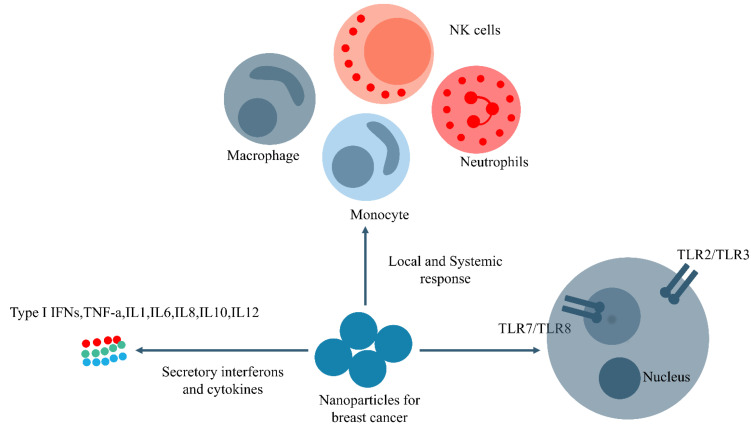
Trained immunity activation for breast cancer therapy using NPs; NPs can provoke various native cells via the induction of TLRs, interleukins, cytokines and chemokines release and the proliferation of innate immune cells via an efficient delivery of components.

## Data Availability

The datasets used and/or analyzed during the current study will be available from the corresponding author on reasonable request.

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
