# Peer review of "Metallic Nanoparticles: Their Potential Role in Breast Cancer Immunotherapy via Trained Immunity Provocation"

_biomedicines, 2023, doi:10.3390/biomedicines11051245_

Round 1

Reviewer 1 Report

Paper is deserved to be published after minor revision.

My comments you can find below:

1. Nanoparticles do not degrade or dissolve easily. Instead, they can accumulate in biological systems and persist for long periods of time, making such nanoparticles a particular problem. How and where do different particles accumulate in the body? and how to solve this problem? Please discuss.

2.Nanoparticles (especially magnetic nanoparticles: IONP, IONP/Au) where used as theranostic agents in bio-medical applications. Synergistic effect of such particles can improve therapeutic selectivity and therapeutic efficiency in general. More discussion should be given.

Author Response

Dear Professors

Editor in Chief/ Reviewers

Biomedicines

Subject: Submission of revised manuscript entitled " Metallic Nanoparticles: their improvement role in Breast Cancer immunotherapy via Trained Immunity provoking

We greatly appreciate the editor and reviewers for their comments and suggestions. We have carefully reviewed the comments and have revised the manuscript accordingly. Our point to point responses are given below. Changes to the manuscript are indicated in blue highlighted sentences/words. We hope that you find our responses satisfactory and that the manuscript is now acceptable for publication. Anyway, we should be grateful if you let us know about our further changes required. Thanks in advance for valuable comments.

Yours sincerely,

*Corresponding author

Reviewer 2 Report

In this review article, the authors revisit the last 10 years of literature about trained immunity elicited by metallic nanoparticles with a focus on breast cancer

It is an interesting topic, and the information provided is extensive, however there is room for improvement in this reviewer’s opinion

MAJOR COMMENTS

Title: (..) their improvement role in breast cancer immunotherapy (..): all data reviewed are preclinical, so a suggestion would be “their potential role”

Conclusion: this section should be re-written with a translational medicine perspective and with an emphasis on realistic potential of MNP to improve breast cancer immunotherapy, and with mention, if any, of ongoing or upcoming clinical trials in this field. This is in part addressed in the paragraph “Future prospects”

MINOR COMMENTS

Abstract: line 19: “No more than a decade, the adaptation of the innate immunity responses against infectious disease and cancer was recognized.” It is not clear what the meaning of this sentence is

Keywords: this reviewer suggests including “trained immunity”

Background:

line 32 “de facto” should be in italics, ideally

line 34: please provide a reference for the sentence “(..) was proposed as trained immunityin 2011.”

line 41 figure 1: please provide a more descriptive and detailed figure legend

line 51 figure 2: same as above

line 79: there is no (table 1) in this manuscript

line 85: please start a new paragraph with the sentence “This paper was a systematic review (..)”

line 92: it should be (Fig. 3) and not (Fig. 4)

Trained immunity cells and compartments

line 97 lymphocytes, if B and T, should be removed

Metallic nanoparticles and immunotherapy of cancers

line 116 “respiration of enzymes” could be rephrased

Nanoparticles for the immunotherapy of breast cancer

line 123 the sentence “IL-6, TNF-a (..) cytokines release, [50-52]” does not seem to make sense

line 141 MRI should be spelled out

line 154 the sentence (..) in models a translation (..) is excluded” does not seem to make sense

line 166 “4T1” should be “4T1”

line 169 “using inducing” is not clear

line 171 “checkpoint obstruction immunotherapy” should be “checkpoint blockade” or “checkpoint inhibition” immunotherapy

line 171 using activating” is not clear

line 235 WKYMV should be spelled out

line 235 “affective” should be “effective”

line 236 “medicine” could be changed into “drug”

line 360 Figure 4 please provide a more descriptive and detailed figure legend

Author Response

(The authors gave the same response as above.)

Reviewer 3 Report

The Review article entitled “Metallic Nanoparticles: their improvement role in Breast 2 Cancer immunotherapy via Trained Immunity provoking” discussed cancer immunotherapy and the application of metal nanoparticles can improve their efficacy. The article has many grammatical and sentence errors, and the language organization needs to be improved. For these reasons, I conclude that the paper should undergo minor revision.

1.      Authors may provide greater details of other cancer therapy with their merits and demerits. How immunotherapy is better than others

2.      Good introduction, Still, it should be improved. To make the introduction more substantial, the author should provide more insight into immunotherapy and its advantages with several updated recent references to substantiate the claim made.

3.      Authors need to provide information about the FDA-approved Major cancer immunotherapeutics. Refer https://doi.org/10.1186/s40824-018-0133-y

4.      How nanoparticles help in increasing the efficacy of cancer immunotherapy. refer https://doi.org/10.1002/smll.201900262.

5.      How nanoparticle helps to improve the deliverables of cancer immunotherapy

6.      Authors need to add sections on the future scopes and challenges in cancer immunotherapy.

7.      There are many grammatical and sentence errors in the article, and the language organization needs to be improved.

Author Response

(The authors gave the same response as above.)

Round 2

Reviewer 2 Report

The authors have considerably improved the manuscript

However, a few minor modifications are suggested

Page 3, line 80 to 83

The short paragraph “Study Design” is not necessary and also repetitive. What is written from line 97 to 104 explains well the study design. This reviewer’s suggestion was simply to start a new line with “This paper was …” without creating a new section of the manuscript 

The abbreviation “MRI” which stands for magnetic resonance imaging should be spelled out where it first appears in the manuscript, i.e., on line 182 and not on line 188

Section 6 “Conclusion” should be “Conclusions”? and “Abbreviation” should be Abbreviations”?

Section “Acknowledgments” should be removed if no need to acknowledge anyone

the last sentence of "Conclusion" line 406-408 is not clear

Author Response

Dear Professors

Editor in Chief/ Reviewers

Biomedicines

Subject: Submission of revised manuscript entitled " Metallic Nanoparticles: their improvement role in Breast Cancer immunotherapy via Trained Immunity provoking

We greatly appreciate the editor and reviewers for their comments and suggestions. We have carefully reviewed the comments and have revised the manuscript accordingly. Our point to point responses are given below. Recent changes to the manuscript are indicated in green highlighted sentences/words. We hope that you find our responses satisfactory and that the manuscript is now acceptable for publication. Anyway, we should be grateful if you let us know about our further changes required. Thanks in advance for valuable comments.

Yours sincerely,

*Corresponding author:

 Abdolmajid Ghasemian

Noncommunicable Diseases Research Center, School of Medicine, Fasa University of Medical Sciences, Fasa, Iran

Tel: +989106806917

Response to the reviewer comments:

Reviewer2:
Page 3, line 80 to 83

Response: It was checked.

The short paragraph “Study Design” is not necessary and also repetitive. What is written from line 97 to 104 explains well the study design. This reviewer’s suggestion was simply to start a new line with “This paper was …” without creating a new section of the manuscript 

Response: It was checked and highlighted with green color.

The abbreviation “MRI” which stands for magnetic resonance imaging should be spelled out where it first appears in the manuscript, i.e., on line 182 and not on line 188

Response: It was corrected.

Section 6 “Conclusion” should be “Conclusions”? and “Abbreviation” should be Abbreviations”?

Response: They were corrected.

Section “Acknowledgments” should be removed if no need to acknowledge anyone

Response: It was removed.

the last sentence of "Conclusion" line 406-408 is not clear

response: It was revised.

With best regards